# Prevalence of Relative Age Effect in Russian Soccer: The Role of Chronological Age and Performance

**DOI:** 10.3390/ijerph16214055

**Published:** 2019-10-23

**Authors:** Eduard Nikolayevich Bezuglov, Pantelis Theodoros Nikolaidis, Vladimir Khaitin, Elvira Usmanova, Anastasiya Luibushkina, Alexey Repetiuk, Zbigniew Waśkiewicz, Dagmara Gerasimuk, Thomas Rosemann, Beat Knechtle

**Affiliations:** 1Department of Sport Medicine and Medical Rehabilitation, Sechenov First Moscow State Medical University, 119435 Moscow, Russia; e.n.bezuglov@gmail.com; 2Exercise Physiology Laboratory, 18450 Nikaia, Greece; pademil@hotmail.com; 3FC Zenit Saint-Petersburg, 197341 Saint-Petersburg, Russia; khaitinvladimir@gmail.com; 4PFC CSKA, 123103 Mocsow, Russia; uscska@mail.ru; 5«Smart Recovery» Clinic, 121552 Moscow, Russia; nastya.lyubuskina@mail.ru; 6FC «Lokomotiv», 107553 Moscow, Russia; repetiuk@fclm.ru; 7Institute of Sport Science, Jerzy Kukuczka Academy of Physical Education, 40-065 Katowice, Poland; z.waskiewicz@awf.katowice.pl; 8Department of Sports Medicine and Medical Rehabilitation, Sechenov University, 119991 Moscow, Russia; 9Department of Sports Training, Jerzy Kukuczka Academy of Physical Education, 40-065 Katowice, Poland; d.gerasimuk@awf.katowice.pl; 10Institute of Primary Care, University of Zurich, 8091 Zurich, Switzerland; thomas.rosemann@usz.ch; 11Medbase St. Gallen Am Vadianplatz, 9001 St. Gallen, Switzerland

**Keywords:** relative age effect, youth football, Russian football, competition in football

## Abstract

The relative age effect (RAE) has been well studied in adolescent and adult soccer players; however, less information has been available about children engaged in regular soccer training and the role of performance. Thus, the aim of the present study was to examine the prevalence of RAE in children and adolescent soccer players, as well as the role of age and performance. Russian soccer players (*n* = 10,446) of various ages, playing positions and performance levels were examined for their date of birth. It was observed that RAE was widespread in Russian soccer teams of all age groups. RAE was most pronounced in children teams of the top tier Russian soccer academies and junior Russia national teams, where the proportions of soccer players born in the first quarter were 43.9% and 39.8%, respectively, whereas those born in the fourth quarter of the year were 7.7% and 6.3%, respectively. In top tier soccer academies, RAE did not vary by age group. In the middle tier soccer academies, RAE was less pronounced. It was still prevalent in the junior teams of the top tier clubs of the Russian Premier League, where 14.3% of the soccer players were born in the fourth quarter of the year compared to 42.9% born in the first quarter of the year. RAE can be observed in the top tier Russian adult teams as well, although it is less pronounced there. In summary, RAE is highly prevalent in Russian children and junior soccer and is associated with the level of competitiveness. At the same time, the proportion of players born in the fourth quarter of the year is higher in adult teams than in junior and youth teams, which is most likely due to the wider selection of players, not limited by their age and place of residence. In junior teams, RAE results in a bias towards selection of players who are more physically mature, whereas children who may be more talented but are less developed due to their younger chronological age tend to be overlooked.

## 1. Introduction

Chronological differences among humans of an age group have been described using the term relative age and their role in human performance has been defined as relative age effect (RAE) [1]. RAE has been highly prevalent in popular competitive sports, such as soccer, hockey, basketball, baseball, and volleyball, and occurred in the majority of age groups and sport clubs regardless of their performance level [2,3,4]. In addition to sports, RAE was also common in education [2,3,4,5]. The first study of RAE in sports was that of Barnsley et al., who observed a bias toward athletes born in the first half of the year in Canadian hockey teams regardless of age group and performance level [6]. Currently, there are many studies devoted to this phenomenon in hockey, where its prevalence continues to remain high [7,8].

RAE has been associated with the popularity, high competitiveness, and early start of systematic training in German soccer [9]. A possible explanation for the high prevalence of RAE was the maturity status theory [8,10,11]. According to this theory, the selection of young soccer players by elite schools was always affected by a bias in favor of boys born in the first half of the year, as they are anthropometrically, cognitively, and physically superior to their peers born in the end of the year. As a result, players who were physically more mature might often be unreasonably considered “gifted”. Equally gifted players who were less well-developed for their age may be demoted to reserve team or dismissed from the team due to their lower physical performance, with no consideration of their future potential. Such players were, therefore, trained by less qualified coaches and received less practice, which resulted in an increasing lack of technical, tactical, and other key skills [8,10,11]. It should be highlighted that anthropometric characteristics in soccer might vary across decades. For instance, the young soccer players were taller in 2000–2015 compared to 1978–1999, which was attributed to the improvement of nutrition and living conditions and to the changes in the selection criteria that favored physical prowess [12].

The talent identification in soccer has been relied on physical performance at a given time, which might complicate the entry into professional sports for late maturing children and children born in the second half of the year [13]. For instance, soccer clubs composed primarily of players born early in the year were more likely to place higher in a tournament of 17-year-old Germans. However, soccer players from clubs with a lower relative age had higher chances to continue their careers in adult teams [14]. On the other hand, it has been reported that the “early-born” members of junior soccer clubs were more likely to be successful in their adult professional careers [15].

RAE was common in many countries with high performance level of soccer, e.g., Germany, England, Spain, Belgium, and the Netherlands. For instance, the number of children born in the first quarter of every calendar year in the junior soccer clubs of Germany and England reached 50% [14]. RAE was highly prevalent among professional young soccer players and amateurs alike and did not depend on player position in youth of the Spanish Premier League clubs and of the largest amateur Spanish academies [16]. Key physical attributes might differ between “early-born” and “late-born” children [17]. In 10-year-old soccer players, the larger half (66%) were born in the first half of the year and “early-born” had longer limbs, higher fat-free body mass, and were more agile than “late-born” children [18]. Moreover, it was observed in junior Belgian soccer players aged 10 to 19 years that the players born in the first quarter of the year were more numerous compared to the players born in the fourth quarter (37.6% and 12.7%, respectively), and there was a tendency for the “early-born” children to be heavier and taller in comparison to the “late-born” children [19,20]. Increased risk of trauma was another factor which limited the professional growth for late maturing and “late-born” children and was related to bone age, volume of received training, and the number of games played in young soccer players aged 9 to 16 years [21].

Although the aforementioned studies enhanced our understanding of RAE, there has been limited information with regards to its variation by performance level, i.e., whether RAE would have different prevalence in high level soccer academies compared to their less competitive counterparts. It would also be interesting to verify whether RAE would be attenuated with age in young soccer players. Furthermore, its prevalence might vary by country and, consequently, there was a need for research in more countries. Russia was a country with high popularity of soccer, and it would be of practical application to examine the prevalence of RAE in this country. Thus, the aim of the present study was to examine the prevalence of RAE in children and adolescent soccer players, along with the role of age and performance.

## 2. Materials and Methods

### 2.1. Study Design and Participants

The present research used a cross-sectional study design to analyze the dates of birth of male soccer players of various ages (minimum age 7 years) living in Russian cities (Moscow, Saint Petersburg, Stavropol, or Krasnodar) and studying in academies of various levels, as well as football players of the main and youth teams of the five leading RPL clubs and national teams of Russia. The selection of Russian teams was based on samples of convenience. To compare the prevalence of RAE in junior and youth soccer with the European teams of similar status, the dates of birth of players accepted into the rosters of Spanish (FC Barcelona, Real Madrid C.F.), English (Manchester United F.C., Chelsea F.C., Tottenham Hotspur F.C., Liverpool F.C.), Croatian (GNK Dinamo Zagreb), Portuguese (S.L. Benfica), and Ukrainian (FC Dinamo Kyiv) clubs participating in the Union of European Football Associations (UEFA) Youth League (*n* = 257).

In addition, the birth dates of all adult football players, who played in all RPL teams at different playing positions from 2001 to 2017 (*n* = 7263), are analyzed. The data were obtained from the web pages of the Russian Premier league (premierliga.ru), the Russian soccer Union (rfs.ru), the Union of European Football Associations (uefa.com), and the International Federation of Association Football (fifa.com), as well as by querying a large Russian sports page championat.com or by directly contacting the administration of the academies. All the players were divided into four groups according to their date of birth, with the first group composed of players born in January, February, and March (“early-born”); the second group of players born in April, May, and June; the third group of players born in July, August, and September; and the fourth group of players born in October, November, and December (“late-born”). For each one of these date of birth groups, the relative sample size was calculated using the formula “100 × number of the date of birth group / total number”. RAE was defined as a higher relative sample size of the first group compared to the other date of birth groups.

The members of the leading Russian soccer academies aged 7 to 17 years were selected (Chertanovo Football Academy, *n* = 233; Football School Lokomotiv, *n* = 255; CSKA Football School, *n* = 269; Zenit Academy, *n* = 202; total *n* = 959). The “top tier academies” were considered the academies of large clubs in which football players were selected from the age of six, they have their own boarding schools for permanent residence of football players, and young football players from other regions come to them. In order to assess the association between the prevalence of RAE and the level of competitiveness in children and youth sports, we performed a comparative analysis of RAE in the teams of the leading soccer academies, junior and youth Russia national soccer teams (*n* = 128), junior teams of the leading clubs of the Russian Premier League (5 teams, *n* = 140), middle tier soccer academies (Kozhany Myach Roman Pavlyuchenko Children’s and Youth Sports School, *n* = 265; Children’s and Youth Sports School of Stavropol, *n* = 281, subdivision of Football School Lokomotiv, *n* = 245, subdivision of Football Academy Chertanovo, *n* = 182, total *n* = 973), as well as in private soccer schools where there is no significant competition (Football Academy Avangard, CFKiS Lobnya, Kvazar; Rodina total *n* = 261).

Regional academies were considered “middle tier academies”, where only residents of the neighboring region were involved in football and there was no boarding school for young players to live permanently. Schools were considered as “private” academies where they accepted everyone who had no medical contraindications to the game of football and where tuition was paid.

### 2.2. Data and Statistical Analyses

To assess the prevalence of RAE in adult Russian soccer, we analyzed the dates of birth of the soccer players accepted into the rosters of the leading Russian Premier League soccer teams that participated in the European club championships in 2018 (FC Lokomotiv, PFC CSKA, FC Spartak, FC Krasnodar, FC Zenit; total *n* = 161). To objectively estimate of the prevalence of RAE in Russian professional club soccer and compare it to the measurements obtained in the teams of other high tier countries, we analyzed the dates of birth of the members of the Russia national soccer team, as well as the top 11 national teams according to the FIFA rankings as of 6 April 2019, and the national team of Germany, winner of the World Championship of 2014 (*n* = 304). Overall, 10,446 dates of birth were analyzed. SPSS Statistics package v.23.0 (IBM, USA) was used for statistical analysis. Chi-square test was used to check the research hypotheses, i.e., whether the relative sample size of the four birth date groups differed with regards to performance level and age. In the Russian general population, 23.8% was born in the first quarter of the year, 23.9% in the second, 26.6% in the third, and 25.7% in the fourth one [20]. Statistical significance was set at alpha = 0.05.

## 3. Results

### 3.1. Performance Level of Soccer Levels

A RAE×performance level of soccer academies was observed (χ^2^ = 95.67, *p* < 0.001), where the “early born” soccer players were more prevalent in the top-tier academy (43.3%) than in the second-tier (25.9%) and private school academy (38.3%). “Late born” soccer players were 7.4%, 16.2%, and 14.9%, respectively (Figure 1).

### 3.2. Age and Performance Level of Soccer Academies

In top tier academies, the ratio of “early-born” to “late-born” players remain similar for all the age groups. A comparison between different age groups did not reveal any statistically significant association between the age and “early-born” to “late-born” ratio (*p* = 0.103). The difference in the ratio of “early-born” to “late-born” players between the three youngest and oldest age groups was not statistically significant (42.3% and 8.7% in the youngest groups; 47.1% and 3% in the oldest groups; *p* = 0.106; Figure 2-left). In the middle tier academies, the number of “late-born” children are comparable to and even higher than the number of “early-born” children, although in several age groups the “early-born” children are still more numerous (Figure 2-right).

Even among the middle tier academies, the one that performs in tournaments with higher level of competition (subdivision of Football School Lokomotiv and subdivision of Football Academy Chertanovo) has more “early-born” players (34.3% and 32,9%, respectively) compared to other middle tier academies (*p* < 0.001), although it is less than in the top tier academies (Figure 3).

The analysis of the birth dates of the members of the junior teams of the leading clubs of the Russian Premier League and Russian national teams of all ages showed that the ratio of “early-born” to “late-born” players was preserved. Junior and youth teams which incorporate the best available players at a particular point in time tend to have the least (only 6.3%) “late-born” players in comparison with all the other analyzed groups. This is most likely connected to the high competitiveness at this tier. This bias can also be noted in the junior teams of the leading clubs of other countries, with England as the only exception, where the “late-born” players are most widely represented and compose 24.1% of the roster. No statistically significant difference in the prevalence of RAE was found between the leading adult and junior Russian teams (*p* = 0.193), where it remained high. It was, however, lower than in the leading international adult teams, where soccer players born in the fourth quarter of the year constituted 19.4% of the roster (Figure 4).

Distribution analysis of the dates of birth of the soccer players showed that the number of ‘late-born’ players in the top tier Russian adult teams is higher than in elite children, junior, and youth teams, although the ‘early-born’ players are still predominant (Figure 5).

### 3.3. Playing Position

Among adult Russian football players who played in all RPL teams from 2001 to 2017, a wide spread of the RAE phenomenon was also revealed (Figure 5), which it found among all playing positions.

“Early born” ranged from 34.7% (midfielders) to 36.1% (forwards), whereas “late born” ranged from 15.5% (defenders) to 17.2% (midfielders) (Figure 6). However, no RAE×playing position was observed (χ^2^ = 3.67, *p* = 0.721).

## 4. Discussion

The main finding of the present study was that RAE was highly prevalent in Russian soccer. It was most pronounced in children and junior teams of top tier academies and national teams, i.e., it was increased with performance level. RAE was less prevalent in academies with lower competitiveness and was absent in schools. Compared to the youth teams, adult teams exhibited a lower prevalence of RAE. This was most likely caused by the option to select the best adult players from a large number of applicants from around the globe. In junior, youth and adult Russian soccer teams, a significant bias toward the “early-born” players were observed in all the analyzed groups. In private soccer schools, where the level of competition is minimal and everyone wishing may attend soccer lessons, there was no statistically significant difference between the proportions of “early-born” and “late-born” players.

These data were consistent with the earlier studies on the prevalence of RAE in junior and adult soccer players of various skill levels worldwide. It has been shown that during the period 2001–2012, RAE was consistently highly prevalent in 10 European soccer championships with no decreasing tendency [22]. In addition to soccer clubs, RAE had high prevalence in junior European national soccer teams composed of players aged 15 to 18 years [23]. The rate of RAE prevalence in elite Norwegian soccer teams was 68% of the players born in the first quarter of the year, and these players received longer play times [24]. Some studies on RAE in soccer highlight the reduction of its prevalence in older groups compared to younger groups [25]. This tendency was not confirmed in the present study, where the prevalence of RAE was very high both in junior Russian national teams and in the top tier Russian Premier League clubs. Mujika et al. [25] analyzed the birth dates of soccer players of different performance originating from the Basque Country. They found that in youth male players RAE correlated with the level of competition.

Among the elite junior soccer players, only 10% were born in the fourth quarter of the year, while at the regional and school levels, they accounted for 21.2% and 22.9% of the players, respectively. That indicates that the level of competition directly influenced the prevalence of RAE. A similar association between the level of competition and the prevalence of RAE was observed in our study. At the same time, in smaller countries where the number of children interested in sports is fewer, coaches are forced to be more flexible, and open-door policies result in a lower prevalence of RAE [26]. Most likely, the inversion of RAE only occurs in African teams. Andrade-Souza found that the African junior national teams were predominantly composed of the players born in the fourth quarter of the year. They tied it to the fact that in these countries the selection for soccer teams is primarily based on emotional, technical, and tactical aspects and not overall physical prowess [27]. Typically, it has been argued that a change would be required in the way the age groups were formed, with the selection window being narrowed and shifted. Several other options have been proposed, among them the calculation of the mean age of the team which should not exceed certain limits, introduction of quotas for “late-born” children, and the modification of the selection process corresponding to a specific set of parameters of physical development calculated for each age group [28,29]. In addition, a database of the key anthropometric and physiological parameters of the leading soccer players of different age would allow to better estimate the potential of the players taking their positions into account [30]. Furthermore, it should be considered that genetically predetermined qualities are of utmost importance in sports such as soccer and may ultimately predetermine the level of success in adult sports [31].

The main reason for the prevalence of RAE in highly competitive sports, such as soccer, is that the trainers choose the players based on their physical maturity and not their talent. A large number of children who are less physically developed but may be just as talented are thus unable to train. Finding a solution to this situation would allow tens of thousands of children and teenagers to practice soccer. That would, in turn, increase the competitiveness and, ultimately, the performance level of professional soccer. The performed observation may influence the existing selection policy in elite soccer and turn the attention of the coaches to the players born at the end of the season.

The disadvantages of the study include the lack of data on the prevalence of RAE in Russian football in previous decades, which allowed us to assess trends in its change over time. In addition, no comparisons were made with ice hockey, another highly competitive sport with an early specialization that was widespread in Russia. In addition, data on the maturation level of participants were not available, and maturation might be a confounding factor on the findings assuming that more matured soccer players would be more “selected”.

The widespread occurrence of RAE in Russian football revealed in the study should lead to the development of a set of measures that can reduce it, thus saving thousands of “late-born” children in football. These measures include: the introduction of quotas for “late-born” players in each team, introduction to the structure of the leading football academies of second teams in each of the ages, consisting of “late-born” young football players, introduction at the first stage of selection at the academy of tests for strength and speed, considering the age of children up to a month, and the introduction of quotas for playing time for all players of leading academies during the competitive season.

Future research should focus on the effectiveness of measures proposed to reduce the prevalence of RAE in highly competitive sports.

## 5. Conclusions

In summary, the present study observed that RAE was highly prevalent in Russian soccer and was associated with age and performance level, and its prevalence remains high at all levels of elite youth football. RAE occurred in top tier Russian adult soccer players at a lesser magnitude than in their younger counterparts, which might be due to a wider pool of soccer players to choose from, without the limitation of the year of birth.

## Figures and Tables

**Figure 1 ijerph-16-04055-f001:**
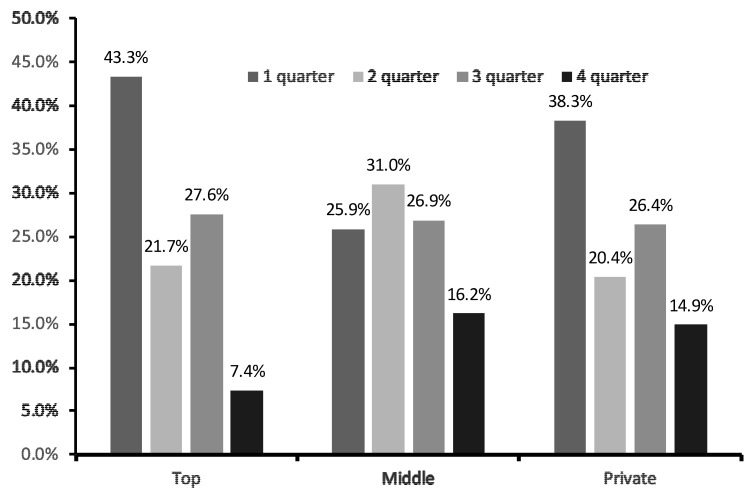
Relative age effect (RAE) by performance group. Top = top-tier soccer academy; Middle = middle-tier soccer academy; Private = academy of private school.

**Figure 2 ijerph-16-04055-f002:**
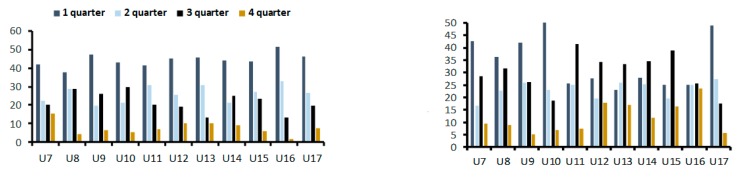
Relative age effect (RAE) by age group in top (left) and middle-tier soccer academies (right). U—indicates age group.

**Figure 3 ijerph-16-04055-f003:**
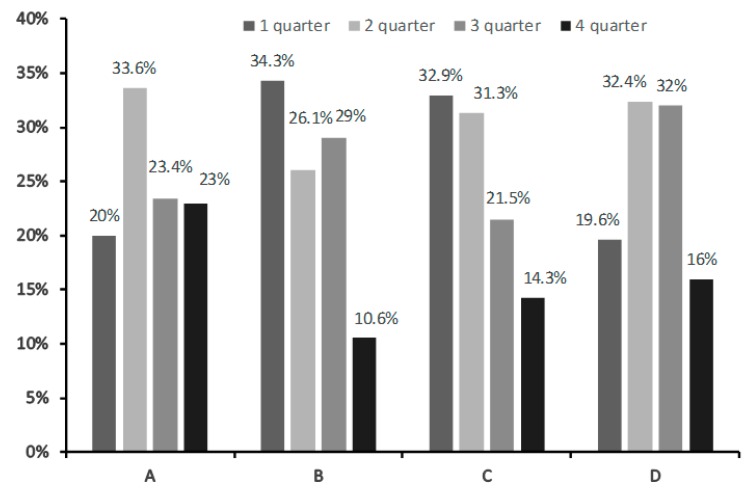
The prevalence of Relative Age Effect in middle tier soccer academies: Kozhany Myach Roman Pavlyuchenko Children’s (**A**), subdivision of Football School Lokomotiv (**B**), subdivision of Football Academy Chertanovo (**C**), and Children’s and Youth Sports School of Stavropol (**D**).

**Figure 4 ijerph-16-04055-f004:**
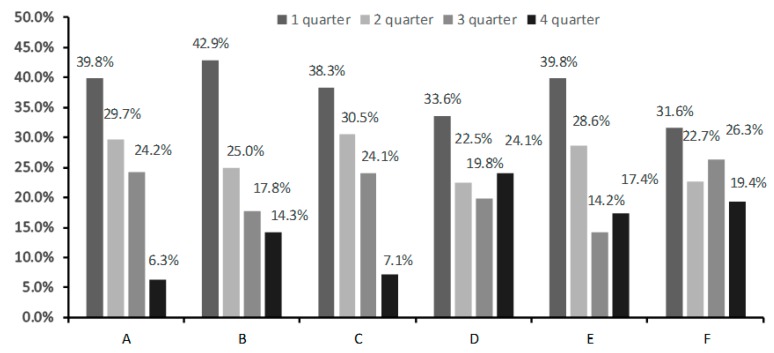
The prevalence of RAE in junior and youth Russia national football teams (**A**), in the leading clubs of the Russian Premier League (**B**), in youth teams of leading non-English clubs participating in the Union of European Football Associations (UEFA) Youth League championships (**C**), in youth teams of leading English clubs participating in the UEFA Youth League championships of 2018–2019 (**D**), in the leading Russian adult football teams (**E**), and in the top 11 countries according to International Federation of Association Football (FIFA) rating as of 6 April 2019, as well as the Germany national team (2014 World champion) and the Russia national team (World championship host) (**F**).

**Figure 5 ijerph-16-04055-f005:**
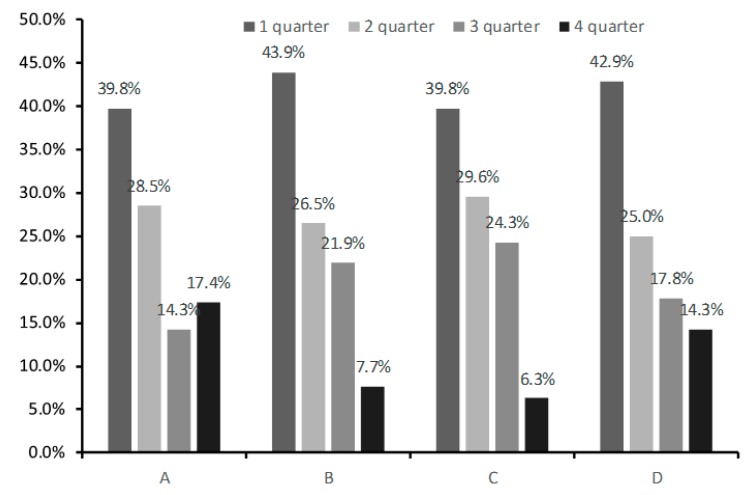
The prevalence of RAE in the leading Russian adult football teams (**A**), top tier football academies (**B**), junior national teams (**C**), and junior teams of the leading Russian football clubs (**D**).

**Figure 6 ijerph-16-04055-f006:**
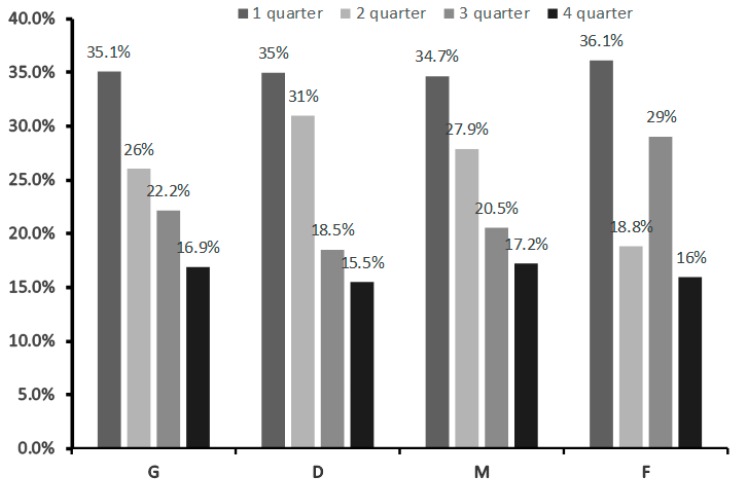
Relative age effect (RAE) by playing position among adult Russian football players who played in all RPL teams from 2001 to 2017 (total *n* = 7263). G = goalkeeper, D = defender, M = midfielder, F = forward.

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
