# Peer review of "Prevalence of Relative Age Effect in Russian Soccer: The Role of Chronological Age and Performance"

_ijerph, 2019, doi:10.3390/ijerph16214055_

Round 1

Reviewer 1 Report

IJERPH manuscript review
Manuscript title: Prevalence of relative age effect in Russian soccer: The role of chronological age and performance
Manuscript number: 582306

This manuscript is a nice analysis of relative age effects (RAE) in
Russian soccer. It summarizes the literature reasonably well although
I would add two notes. First, the literature on RAE in hockey is
extensive, and I would encourage the authors to build up this part of
their literature review. Second. The manuscript states (page 1, line
51) that there is no evidence of relative age effects in American
football, and a new paper by Heneghan and Herron (Journal of
Quantitative Analysis of Sports, September, 2019) shows otherwise.

The empirical analysis consists of chi-squared tests, which is typical
for the RAE literature. I do not fully understand the comments on page
3, lines 114-116, regarding usage of early and late. Some figure in
the manuscript show early, other, and late, and others do not (see
below on this point). Regardless, the manuscript should not just
compare early and late months as this induces a non-conservative bias
into the manuscript's tests. The reason for this is that early births
can differ from late births in a way that is consistent with RAE, but
other month births may not be consistent with RAEs. In other to make
sure this is not the case, the manuscript needs to update its
chi-squared tests so that all player births are considered, not just
early and late one. The RAE literature tends to group births into
quarters or months.

Here are some comments on the manuscript figures.

Figure 1: rescale vertical axis to use percentages.

Figure 2: same issue about scaling. Moreover, young (U7 - U10) and
middle tier group of teams has very small numbers. The standard
chi-squared approximation will not be valid with these small sample
sizes. So, this part of the manuscript should probably be dropped.
Or, additional teams need to be found. The same issue for U17.

Figure 3-5. I would remove all pie charts. They are hard to read and
less useful than the other bar charts in the manuscript.

Figure 6. There appears to be no other group in this figure, and this
needs to be addressed.

Lastly, I cannot tell what the null age distribution is throughout the
manuscript. To establish evidence for an RAE in Russian soccer, it is
not enough to note that there are more relatively older players than
relatively young players. It could be that patterns of Russian births
across months are not uniform.

I am recommending an R&R for this manuscript and look forward to
reading a revision.

Author Response

This manuscript is a nice analysis of relative age effects (RAE) in Russian soccer. It summarizes the literature reasonably well although I would add two notes.

First, the literature on RAE in hockey is extensive, and I would encourage the authors to build up this part of their literature review.

Answer: We agree with the expert reviewer and changed as requested.

Second. The manuscript states (page 1, line 51) that there is no evidence of relative age effects in American football, and a new paper by Heneghan and Herron (Journal of Quantitative Analysis of Sports, September, 2019) shows otherwise.

Answer: We agree with the expert reviewer. We added the reference in the text.

The empirical analysis consists of chi-squared tests, which is typical for the RAE literature. I do not fully understand the comments on page 3, lines 114-116, regarding usage of early and late. Some figure in the manuscript show early, other, and late, and others do not (see below on this point). Regardless, the manuscript should not just compare early and late months as this induces a non-conservative bias into the manuscript's tests. The reason for this is that early births can differ from late births in a way that is consistent with RAE, but other month births may not be consistent with RAEs.

In other to make sure this is not the case, the manuscript needs to update its chi-squared tests so that all player births are considered, not just early and late one. The RAE literature tends to into
quarters or months.

Answer: We agree with the expert reviewer. We separate now groups by quarter.

Here are some comments on the manuscript figures.

Figure 1: rescale vertical axis to use percentages.

Answer: We agree with the expert reviewer and changed as requested.

Figure 2: same issue about scaling.

Answer: We agree with the expert reviewer and changed as requested.

Moreover, young (U7 - U10) and middle tier group of teams has very small numbers. The standard chi-squared approximation will not be valid with these small sample sizes. So, this part of the manuscript should probably be dropped. Or,. The same issue for U17.

Answer: We agree with the expert reviewer and updated with additional teams.

Figure 3-5. I would remove all pie charts. They are hard to read and lss useful than the other bar charts in the manuscript.

Answer: We changed the pie charts to bar charts.

Figure 6. There appears to be no other group in this figure, and this needs to be addressed.

Answer: We agree with the expert reviewer and changed as requested.

Lastly, I cannot tell what the null age distribution is throughout the manuscript. To establish evidence for an RAE in Russian soccer, it is not enough to note that there are more relatively older players than relatively young players. It could be that patterns of Russian births across months are not uniform.

Answer: We agree with the expert reviewer and added this information in the methods.

Reviewer 2 Report

The authors submitted a manuscript about the prevalence of the relative age of performance in Russian players. I think the idea of the manuscript has been previously described in the literature. The authors presented similar results to those already published except that their population has not been studied before. Several comments and suggestions should be considered before the manuscript can be approved for publication.

Introduction

The introduction seems to be adequate for the content of the manuscript. It is clear and brief with updated information. However, the objective does not describe the content of the manuscript and does not involve the steps of statistical analysis.

Materials and Methods

The methodology section has no structure (information is unclear). It is recommended to divide into sub-sections to improve understanding. In addition, more information must be described.

For example:

What study design was used?

Which sampling method was used for the selection?

How was the prevalence (equation) calculated?

You describe that the sample was: In order to get a clearer idea, only the birth dates of the ‘early-born’ and ‘late-born’ players were used for statistical analysis. However, in the figures, the sampling is divided into three categories. Please, you should justify this subdivision.

Statistical analysis should be described in greater depth. I cannot perform an in-depth review of the statistical analysis because the information shown by the authors is not clear.

Results

In the results, you use different concepts. For example "top-tier academy", which soccer academies were included for this analysis or classification? What were their ages at each level? You should standardize the description of the methodology and the results.

Age can be a confounding factor in your results (different times of maturation). How did you control this variable? Results divided by age are not shown.

Regarding the playing position, it has not been described in which sample was calculated

Discusion

The first paragraph of the discussion is very well written and I think it helps to summarize the results very clearly. This is a brief discussion that is relevant to the objectives of the manuscript. However, it must be supplemented by the limitations and practical applications.

Conclusion

The following paragraph: "The main reason for the prevalence of EOV in highly competitive sports such as soccer is that the trainers choose the players based on their physical maturity and not their talent. A large number of children who are less physically developed but may be just as talented are thus unable to train. Finding a solution to this problem would allow tens of thousands of children and teenagers to practice soccer. That would, in turn, increase the competitiveness and, ultimately, the performance level of professional soccer. The performed observation may influence the existing selection policy in elite soccer and turn the attention of the coaches to the players born at the end of the season”.

I think it should not be included in the conclusions section, because it has not been an objective of the authors. It should be included at the end of the discussion, including references to justify these arguments. It could be part of the practical applications requested for the discussion.

Minor:

Line 73, page 2: Please, remove the double comma

Line 82, page 2: Please replace "the majority" with another word which most closely matches 66%.

The titles of the figures should be eliminated. The information must be included in the figure caption.

In figure 2, what does U mean? All the abbreviations should be shown in the figure legend.

Author Response

The authors submitted a manuscript about the prevalence of the relative age of performance in Russian players. I think the idea of the manuscript has been previously described in the literature. The authors presented similar results to those already published except that their population has not been studied before. Several comments and suggestions should be considered before the manuscript can be approved for publication.

The introduction seems to be adequate for the content of the manuscript. It is clear and brief with updated information. However, the objective does not describe the content of the manuscript and does not involve the steps of statistical analysis.

Answer: We agree with the expert reviewer and added this aspect in the end of introduction before aims (“i.e., whether RAE would have different prevalence in high level soccer academies compared to their less competitive counterparts. It would be also interested to verify whether RAE would be attenuated with age in young soccer players”).

Materials and Methods

The methodology section has no structure (information is unclear). It is recommended to divide into sub-sections to improve understanding. In addition, more information must be described.

For example:

What study design was used?

Which sampling method was used for the selection?

How was the prevalence (equation) calculated?

Answer: We agree with the expert reviewer and added subheadings 2.1. and 2.2. to improve the structure. In addition, we added information about the study design (“The present research used a cross-sectional study design to analyze the...”), the sampling method (“The selection of Russian teams was based on samples of convenience.”) and the calculation of prevalence (“For one of these date of birth groups were, the relative sample size was calculated using the formula ‘100 × number of the date of birth group / total number’. RAE was defined as a higher relative sample size of the first group compared to the other date of birth groups.”) as requested.

You describe that the sample was: In order to get a clearer idea, only the birth dates of the ‘early-born’ and ‘late-born’ players were used for statistical analysis. However, in the figures, the sampling is divided into three categories. Please, you should justify this subdivision.

Answer: We agree with the reviewer, the separation by date of birth has been revised and, now, it is divided into four quarters.

Statistical analysis should be described in greater depth. I cannot perform an in-depth review of the statistical analysis because the information shown by the authors is not clear.

Answer: We agree with the reviewer and revised this part (“research hypotheses, i.e. whether the relative sample size of the four birth date groups differed with regards to performance level and age. Statistical significance was set at alpha=0.05.”)

Results

In the results, you use different concepts. For example, "top-tier academy", which soccer academies were included for this analysis or classification? What were their ages at each level? You should standardize the description of the methodology and the results.

Answer: We agree with the expert reviewer. We made the necessary rectification regarding the level soccer academies, which were included for this analysis

Age can be a confounding factor in your results (different times of maturation). How did you control this variable? Results divided by age are not shown.

Answer: We agree with the expert reviewer and added the aspect of maturation in the limitations of the study (“In addition, data on the maturation level of participants were not available, and maturation might be a confounding factor on the findings assuming that more matured soccer players would be more ‘selected’.”). The results by age are presented in fig.2.

Regarding the playing position, it has not been described in which sample was calculated

Answer: We agree with the expert reviewer. We explain now in which sample the playing position was calculated

Discussion

The first paragraph of the discussion is very well written and I think it helps to summarize the results very clearly. This is a brief discussion that is relevant to the objectives of the manuscript. However, it must be supplemented by the limitations and practical applications.

Answer: We agree with the expert reviewer. The limitations and practical applications are now included.

Conclusion

The following paragraph: "The main reason for the prevalence of EOV in highly competitive sports such as soccer is that the trainers choose the players based on their physical maturity and not their talent. A large number of children who are less physically developed but may be just as talented are thus unable to train. Finding a solution to this problem would allow tens of thousands of children and teenagers to practice soccer. That would, in turn, increase the competitiveness and, ultimately, the performance level of professional soccer. The performed observation may influence the existing selection policy in elite soccer and turn the attention of the coaches to the players born at the end of the season”.

I think it should not be included in the conclusions section, because it has not been an objective of the authors. It should be included at the end of the discussion, including references to justify these arguments. It could be part of the practical applications requested for the discussion.

Answer: We agree with the expert reviewer and changed as suggested.

Minor:

Line 73, page 2: Please, remove the double comma

Answer: We agree with the expert reviewer. We deleted the double comma.

Line 82, page 2: Please replace "the majority" with another word which most closely matches 66%.

Answer: We agree with the expert reviewer. We replaced "the majority" with «larger half».

The titles of the figures should be eliminated. The information must be included in the figure caption.

Answer: We agree with the expert reviewer. We eliminated the titles of the figures.

In figure 2, what does U mean? All the abbreviations should be shown in the figure legend.

Answer: We agree with the expert reviewer. We clarified the abbreviations «U».

Round 2

Reviewer 1 Report

I have read the new draft and like it.  I confirmed the author's updates and agree that they have improved the manuscript.  The only update that was not complete is this: 

Second. The manuscript states (page 1, line 51) that there is no evidence of relative age effects in American football, and a new paper by Heneghan and Herron (Journal of Quantitative Analysis of Sports, September, 2019) shows otherwise.

Answer: We agree with the expert reviewer. We added the reference in the text.

Two things.

1.  I would add this citation to the material on hockey: https://journals.plos.org/plosone/article?id=10.1371/journal.pone.0182827

2.  The Heneghan and Herron paper is not cited in the manuscript, despite the fact that the report says that it is. 

Both of these changes should be very easy.

The other comment I would make is that the new material on p. 8 is a bit speculative.  We do not know if the sentence starting with, "A large number of children..."  I share the author's concern, but I think it should be made clear in this material that the author is thinking broadly about sport and society.  Moreover, I am not sure if the word "problem" is the right one in this context.  Should people be working against RAEs, as the author suggests around line 284?  I do not object to this material, but I do think that the author should caveat the concerns here.  Maybe the existence of RAEs are normatively troubling, and maybe they are not.

Author Response

Second. The manuscript states (page 1, line 51) that there is no evidence of relative age effects in American football, and a new paper by Heneghan and Herron (Journal of Quantitative Analysis of Sports, September, 2019) shows otherwise.

Answer: We agree with the expert reviewer. We added the reference in the text.

Two things.

I would add this citation to the material on hockey: https://journals.plos.org/plosone/article?id=10.1371/journal.pone.0182827 The Heneghan and Herron paper is not cited in the manuscript, despite the fact that the report says that it is.

Answer: Yes, we agree with the expert reviewer, this is our mistake. We added both papers in the text.

Both of these changes should be very easy.

The other comment I would make is that the new material on p. 8 is a bit speculative.  We do not know if the sentence starting with, "A large number of children..."  I share the author's concern, but I think it should be made clear in this material that the author is thinking broadly about sport and society.  Moreover, I am not sure if the word "problem" is the right one in this context.  Should people be working against RAEs, as the author suggests around line 284?  I do not object to this material, but I do think that the author should caveat the concerns here.  Maybe the existence of RAEs are normatively troubling, and maybe they are not.

Answer: We we agree with the expert reviewer. We made some adjustments in the text

Reviewer 2 Report

The authors have responded to all of my comments and have revised the manuscript according to my suggestions

Author Response

than you for your comments